# Synergistic Inhibition Effect of Chitosan and L-Cysteine for the Protection of Copper-Based Alloys against Atmospheric Chloride-Induced Indoor Corrosion

**DOI:** 10.3390/ijms221910321

**Published:** 2021-09-25

**Authors:** Elena Messina, Chiara Giuliani, Marianna Pascucci, Cristina Riccucci, Maria Paola Staccioli, Monica Albini, Gabriella Di Carlo

**Affiliations:** 1Istituto per lo Studio dei Materiali Nanostrutturati-Consiglio Nazionale delle Ricerche, Area della Ricerca CNR-Montelibretti, via Salaria km 29.3, 00015 Monterotondo, RM, Italy; chiara.giuliani@enea.it (C.G.); marianna.pascucci@cnr.it (M.P.); cristina.riccucci@cnr.it (C.R.); mariapaola.staccioli@ismn.cnr.it (M.P.S.); monica.albini@cnr.it (M.A.); 2Division Sustainable Materials, ENEA, C.R. Casaccia, Via Anguillarese 301, S. M. Di Galeria, 00123 Rome, Italy

**Keywords:** sustainable corrosion protection, active materials for conservation of Cu-based alloys, chitosan protective coatings, green inhibitors, FTIR surface studies

## Abstract

The protection of metals from atmospheric corrosion is a task of primary importance for many applications and many different products have been used, sometimes being toxic and harmful for health and the environment. In order to overcome drawbacks due to toxicity of the corrosion inhibitors and harmful organic solvents and to ensure long-lasting protection, new organic compounds have been proposed and their corrosion inhibition properties have been investigated. In this work, we describe the use of a new environment-friendly anticorrosive coating that takes advantage of the synergism between an eco-friendly bio-polymer matrix and an amino acid. The corrosion inhibition of a largely used Copper-based (Cu-based) alloy against the chloride-induced indoor atmospheric attack was studied using chitosan (CH) as a biopolymer and l-Cysteine (Cy) as an amino acid. To evaluate the protective efficacy of the coatings, tailored accelerated corrosion tests were carried out on bare and coated Cu-based alloys, further, the nature of the protective film formed on the Cu-based alloy surface was analyzed by Fourier-transformed infrared spectroscopy (FTIR) while the surface modifications due to the corrosion treatments were investigated by optical microscopy (OM). The evaluation tests reveal that the Chitosan/l-Cysteine (CH/Cy) coatings exhibit good anti-corrosion properties against chloride attack whose efficiency increases with a minimum amount of Cy of 0.25 mg/mL.

## 1. Introduction

Numerous studies have been carried out to investigate metal corrosion phenomena occurring in different environmental conditions and various methods and inhibiting materials, sometimes toxic, have been proposed and implemented to prevent metal dissolution. In the last decades, a large effort has been devoted to replace the toxic or low-toxic organic and inorganic inhibitors with green compounds characterized by a good anti-corrosion efficiency and which do not pose health or ecological hazards [1,2,3,4,5,6].

These studies have shown that molecules containing both nitrogen and sulfur atoms such as the mercapto-functionalized thiadiazole derivatives [5,6,7,8,9] and amino acids [10], among others, provide better inhibition efficiency compared to organic inhibitors containing only sulfur or nitrogen. In particular, amino acids have been considered possible candidates because they are non-toxic, water-soluble and biodegradable materials, therefore, they have been proposed for the corrosion prevention of metals and alloys [11,12,13,14,15,16,17,18,19,20] occurring in different aggressive media. These studies have revealed that the amino acids are adsorbed on the surface through the heteroatoms, forming a blocking barrier on the metal surface thus decreasing the corrosion rate.

The interesting anti-corrosion property of amino acids has been also reported by Moretti et al. [21], who reported the good inhibiting action of tryptophan (Trp) in the short-time exposure of pure copper to aerated 0.5 M sulfuric acid. By using potentiodynamic and spectrophotometric techniques, the positive effect of Trp was evidenced especially at the highest concentrations and temperature as well as after six months of specimen immersion in concentrated Trp solution (10^−2^ M) in H_2_SO_4_ 0.5 M.

Ashassi-Sorkhabi et al. [19] have explained the successful application of amino acids as a corrosion inhibitor for aluminum in mixed acid solution by pointing out the role of S atom in increasing the interaction of the molecule with the metal surface. The results have also revealed the interesting behavior of the glycine in neutral chloride solutions whose inhibition efficiency is about 85% at a very low concentration of amino acids (0.1 mM).

Our attention has been therefore attracted by the Cy considering that the thiol group of Cy is reactive and has a high affinity for copper and its alloys and could be efficient as a corrosion inhibitor for the presence of amino group [–NH_2_] [22]. It is worth noting that the effect of Cy on the dissolution of copper in 0.6 M NaCl and 1.0 M HCl was previously examined by Ismail [23] revealing that the mechanism of the corrosion inhibition is based on the adsorption of the Cy on the active corrosion sites and their anticorrosion efficiency is comparable with those of toxic inhibitors.

Results revealed that cysteine is a good inhibitor for copper corrosion at about 18 mM in HCl solution. A further increase in cysteine concentration showed a slight decrease in its performance.

Cy was selected since in addition to the amino group, it also contains the –SH groups which have a strong affinity to copper while CH was selected due to its good film-forming ability, water-solubility, superior adhesion to metallic surfaces and transparency.

In this scenario, the use of CH that acts as a barrier layer will permit an opportunity for lowering Cy concentration further (up to 2 mM).

The CH/Cy coating properties were optimized paying attention to the requirements necessary for their application in the field of cultural heritage in particular for the work of art characterized by a shiny surface finish typically of the modern works of art.

Similarly, in the field of polymer-based coatings, many efforts have been devoted to the study of the corrosion inhibition behavior of metals by CH and its derivatives due to specific properties such as good film-forming ability, superior adhesion to metallic surfaces, transparency and versatility associated with the easiness of chemical functionalization [24,25,26,27].

These capabilities have been confirmed by our recent experiments that have allowed us to define a valid strategy for the protection of metal Cu-based substrates based on the use of “active” coatings consisting of a passive polymer matrix (CH) loaded with a reduced amount of benzotriazole and mercaptobenzothiazole [28,29].

In order to overcome drawbacks due to the toxicity of these inhibitors and the harmful organic solvents, we have combined the above-cited positive properties of CH with the anti-corrosive properties of Cy. This approach was adopted to prevent an atmospheric chloride attack against Cu-based alloys with a mirror-like surface finish typical of modern artwork.

Conventional electrochemical measurements with immersion in water solutions cannot be used to quantify their protective efficacy since they would cause coating dissolution. The protective efficacy of the coatings was investigated by means of accelerated corrosion treatments in the presence of aggressive species and proved able to promote degradation processes and rapidly provide information. In order to quantify the ability of coatings to prevent degradation processes, an image analysis protocol was exploited, which was also used in previous works [28,29].

The present study proves that polymer coatings containing amino acids ensure long-lasting and reliable protection against corrosion and are thus suitable for cultural heritage preservation purposes.

## 2. Results

Chitosan solutions were prepared by dissolving 0.5 wt/vol% of purified chitosan in aqueous 0.05 M D-(+)-gluconic δ-lactone (GDL) solution as in the previous work [28]. The concentrations of the two acids was selected in order to obtain a solution with a pH~3.5.

The corrosion inhibitor Cysteine (two different percentages with respect to chitosan solution) was added to aqueous Chitosan solutions with GDL. All the coatings were prepared by drop-casting 60 μL chitosan-based solutions onto bronze disks with a diameter of 2.5 cm and by subsequent drying at room temperature.

In order to evaluate the role of Cy and CH to protect Cu-based alloys against atmospheric chloride attack, accelerated corrosion tests were conducted by varying the CH/Cy ratio.

Different chitosan formulations were used: pure-chitosan and chitosan with two different cysteine concentrations.

In Figure 1, the table shows the general appearance and the optical images of the bare disk and the same disk coated with different chitosan formulations. The images reveal that the coating with pure-chitosan and chitosan formulation with a CH/Cy 0.5 mg/mL show up colorless, more transparent and uniform than the coating prepared by using a great amount of Cy (CH/Cy 10), meeting the aesthetic requirements for the protection of cultural heritage. The advantage of CHI/Cy 0.5 in respect to pure-chitosan is the presence of Cy molecules adsorbed at the metal surface that could be highly effective in preventing corrosion.

However, the excessive Cy amount (10 mg/mL i.e., 85 mM) brings about a dramatic reduction of coating anticorrosion ability, since it promotes bubble formation on its surface (see Figure 1).

Ismail [23] reported a maximum inhibition efficiency of Cy as an inhibitor to almost 16 mM, while further increase in amino acid concentration could result in a slight decrease in performance.

To clarify the properties of active coatings at different Cy amounts, FTIR spectroscopy has been used to determine the diverse coating reactivity.

The FTIR spectra of CH/Cy 10 mg/mL and CH/Cy 0.5 mg/mL films deposited onto Cu-based alloys before acid treatment are shown in Figure 2.

The FT-IR spectra exhibit the characteristic bands of amino acid and polymer matrix and suggest that although the chemical composition was identical, the frequencies were slightly different from each other [30,31].

The spectrum of CH/Cy 10 mg/mL exhibits a weak band near 2550 cm^−1^ and a more intense band at 940 cm^−1^ (pink stars in Figure 2) confirming the presence of S-H group in the active coating. Strong characteristic absorptions at 1405 cm^−1^ (-C-OH bending carboxyl group) and 1045 cm^−1^ (-C-O stretching carboxyl group) suggest that the CH with aldehyde group end reacted with Cy (black stars in Figure 2) [32]. These spectral data strongly indicated that cysteine residues coupled with the aldehyde groups on the chitosan ends deteriorated the active coating [33]. The band at 1040 cm^−1^ was attributed to S=O stretch vibration due to oxidation of cysteine to cysteic acid and that might excuse its yellowing. [34,35]. Oxidation of cysteine results in the formation of disulfide bonds between the sulfur molecules or to further oxidation in products of the thiol group. A distinction between cysteine and dimeric amino acid connected via an S-S-bond is not possible by FTIR spectroscopy alone, which is not IR active due to a missing dipole momentum.

On the other hand, the peak at 2550 cm^−1^ is not detectable in CH/Cy 0.5 mg/mL, in agreement with absorption of amino acids molecules towards the alloy surface and the formation of a protective inhibitor layer at the polymer-alloy interface (see Figure 2 red line) [14].

Thus, the intensity of the thiol signal decreased proportionally to a decrease in Cy amount in the active coating, though the amino acid amount depends on a diverse interaction with polymer and as a result of the alloy surface. For each coating, the small peak at ~880 cm^−1^ was assignable to a wagging (the C-H bending out of the plane of the ring) of the saccharide structure of chitosan [36].

Since the mechanism of the corrosion is an inhibition process based on the adsorption of the amino acid on the active corrosion sites CH/Cy 0.5 mg/mL (i.e., 4 mM) was selected as the most promising candidate for the protection of a copper-based alloy.

To evaluate the efficacy of the coatings, accelerated corrosion tests carried out in the presence of acid water vapors reproduced in a quite realistic way the actual aging environment.

Strong inorganic HCl was used since is it well known that chloride ions promote corrosion processes in copper-based alloys [37].

The results of the image analysis, performed to estimate the percentage of corroded surface, were reported in Figure 3. The occurrence of corrosion processes clearly observed in the bare disk was used as reference. In the case of bronze disks coated with pure chitosan, the corrosion treatments affect the coating transparency and removability and lead to the modification of the alloy substrate. The alloy substrate after 180 min showing some small bubbles formed on the surface could be an indicator of a chemical reaction taking place.

As shown, CHI-Cy 0.5 films maintain their homogeneity after the corrosion test. Optical image at low magnification clearly shows the nonoccurrence of degradation processes in the alloy such as the absence of damages onto the coated alloy surfaces.

To address this issue, pure chitosan coating and CHI-Cy 0.5 investigated also FT-IR spectroscopy after the accelerated corrosion test (Figure 4).

In pure-chitosan, the small peak at 880 cm^−1^ of the saccharide in structure of chitosan [36] disappears after the thermal treatment with acid vapors, thus supporting the hypothesis of polymer degradation probably due to the hydrolysis of the pyranose ring catalyzed by the acidic environment.

This evidence suggests that the presence of inhibitors prevents modifications both of the coating and of the alloy substrates, slowing polymer degradation.

In previous works [28,29], we showed that the polymer matrix probably acts as an inhibitor reservoir and contributes to the formation of a barrier layer, thus improving the protective properties.

The protective efficacy of CHI/Cy 0.5 coatings was further investigated by prolonging the accelerated corrosion treatments and observing the occurrence of surface modifications by optical microscopy.

The efficiency of chitosan-based coating was evaluated after accelerated corrosion treatments for 250 min (see Figure 5).

When prolonging the accelerated corrosion treatments, significant differences in the protective efficacy of the coating emerge and a considerable amount of corrosion products (mainly copper oxide and copper hydroxychlorides) can be recognized on the metal surface after accelerated corrosion treatment. Indeed, after 250 min, the OM images revealed that the size and number of the bubbles increased, as well as the corrosion phenomena, probably due to the low adherence of the active coating. Consequently, the coating showed significantly lower transparency in respect to untreated coating.

Following 250 min, the Cu-based alloys CHI/Cy 0.5 mg/mL is still not corroded even though the formation of bubbles in the coating adversely affects the use of the coating for the conservation of metal cultural heritage due to poor aesthetic features.

In this scenario, a new approach to enhance the amino acid performance was probed. First, the alloy bare was immersed in aqueous solution (pH = 7) of Cy (0.02 mg/mL) for 30 min to promote the interaction between molecule and the Cu-based alloy surface.

In order to understand the inhibition mechanism and the interaction with alloy surface, FTIR analyses were performed (see Figure 6). The spectrum of neat Cy has been showed by comparison.

The spectrum exhibits that a thin layer of the amino acid molecule coated the surface, confirmed by disappearance of thiol vibration (blue line).

This observation was further strengthened when one considers the shift at low frequency in the vibration stretching of carbonyl (COO-) in the spectrum of Cy/alloy that confirms the interaction between acid group and other cysteine monomers, possibly via hydrogen bonding. In the same way, other shifts in frequencies of the other group may be because of Cy binding on the alloy surface.

Even so, it seems that the presence of negative ions such as chloride ions, which usually are characterized by strong absorbability on the metal surface, may enhance the adsorption of the amino acid [38,39,40].

The combination of this phenomena and the use of polymer as an inhibitor reservoir permits an opportunity for lowering amino acid concentration further (up to 2 mM i.e., 0.25 mg/mL). The active coating CH/Cy 0.25 mg/mL was deposited by drop casting onto a copper alloy already covered by amino acid and the efficiency in the presence of acid vapor solution was evaluated. Accelerated corrosion tests were carried out by coated alloy in the presence of acid vapors for 180 min and 250 min.

The obtained results confirm that chitosan/inhibitor coatings noticeably protect the bronze surface from corrosion and their efficacy is much more pronounced in the presence minimal amount of amino acid.

Figure 7 shows the evolution of the surface of the CH/Cy 0.25 mg/mL specimen surface with time of treatment in the acid solution. In particular, after 180 min of accelerated corrosion treatment, the surface alloy occurs to be not very corroded (see Figure 7a). When prolonging the accelerated corrosion treatments, significant differences in the protective efficacy of the CHI-Cy 0.5 and CHI-Cy 0.25 coatings emerge.

After 250 min, the Cu-based alloys CHI/Cy 0.25 mg/mL are still fully not corroded and even the formation of bubbles in the coating is not observed (see Figure 7b).

Therefore, the surface did not show changes with degradation time, thanks to the presence of the thin layer of Cy onto the alloy surface as well as the smaller inhibitor quantity into the polymer solution.

The main reason for such difference can be ascribed to the different uptake of protective corrosion inhibitors exclusively on the alloy surface.

It is clear that the presence of cysteine in both solutions decreases the corrosion ac-tivity, though CH/Cy 0.5 shows a considerable decrease in the inhibition efficiencies with the evident formation of more corrosion products after 250 min treatment (Figure 7b).

Based on these findings, chitosan/cysteine coatings results are interesting and promising for the corrosion protection of copper.

The obtained results confirm that chitosan/inhibitor coatings noticeably protect the bronze surface from corrosion and their efficacy is much more pronounced in sample CH/Cy 0.25 mg/mL.

Furthermore, the results of the transparency measurements suggest that polymer amino acid coatings are able to preserve the aesthetic properties of the metal substrate, considering that these new formulations are more sustainable and safe than commercial products typically used for the conservation of bronze works of art.

## 3. Materials and Methods

### 3.1. Reagent and Materials

The Cu-based alloy used as substrate has been industrially produced via continuous casting, its chemical composition is as follows: Sn 5 weight percent (hereafter wt%), Pb 5 wt%, Zn 5 wt%, Cu balance (85 wt%). This chemical composition was selected being representative of modern works of art [28,29].

Chitosan (medium molecular weight, viscosity of 200–800 cP, 75–85 % deacetylated), D-(+)-gluconic δ-lactone (purity ≥ 99.9%), L-cysteine (analytical grade), Ethanol (EtOH, purity ≥ 99.8%) and water Cromasolv plus for HPLC were commercially acquired from Sigma Aldrich.

The as purchased CH has been first carefully washed in boiling water for 60 min, then filtered, thoroughly washed with distilled water to remove the remaining impurities, and finally dried under vacuum for 12 h at room temperature. The CH solution used to produce the thin coatings was prepared by dissolving 0.5 wt%/vol% in aqueous 0.05 M gluconic δ-lactone solution with a pH of ca 3.5. A solution of 1M NaOH was used to adjust the pH value of the polymer solution at about 5.5, according to the scheduled activities different amounts of cysteine was then added to aqueous chitosan solution. (0.5 mg/mL and 10 mg/mL respectively).

Furthermore, the protective coatings have been deposited by immersing first the Cu-based alloy substrates in an aqueous solution of Cy (0.02 mg/mL) for 30 min, thus attempting to promote the interaction between the molecule and the metal surface and then, by depositing on this surface a thin coating of CH-Cy (0.25 mg/mL).

All the coatings were prepared by drop-casting 60 μL of chitosan cysteine-based solutions onto bronze disks with a diameter of 2.5 cm and by subsequent drying at room temperature. Removability tests were carried by using tissue paper soaked in water or ethanol.

Accelerated corrosion experiments [28,29] were carried out by exposing the coated Cu-based alloy to HCl vapors generated by a solution of HCl 1M in a closed glass vessel placed in an electrically heated oven at 50 °C for different treatment times. These strongly enhanced corrosive conditions had better simulate an indoor atmospheric chloride attack and cause an accelerated degradation similar to what really happens. It is worth pointing out that chlorides are considered among the most harmful corroding agents of archaeological or ancient metal artifacts [41,42]. Furthermore, we have used this test [28,29] as an alternative method of the electrochemical tests carried out by immersion in acidic solutions. Removal tests to evaluate one of the requirements for the use of these coatings for the protection of bronze artworks was carried by using tissue paper soaked in water or ethanol.

Five independent measurements were carried out for each type of coating investigated to take into account the variability due to the surface of the bronze disks. In such a way, the reproducibility of the CH/Cy 0.25 mg/mL was verified as the best coating.

### 3.2. Optical Microscopy (OM)

The OM characterization was performed to observe modifications occurring at the surface of the Cu-based alloy surface after accelerated corrosion tests. The OM images were acquired without removing the coatings and therefore, observing only the modification induced by the accelerated degradation treatments. OM morphological investigations were performed by means of a Leica MEF IV inverted optical microscope with bright-field, dark-field, polarized light, polarization contrast and differential interference contrast. To record the digital optical images, a high-resolution 420 CCD Camera was used.

### 3.3. Attenuated Total Reflectance Fourier-Transform Infrared Spectroscopy (ATR-FTIR)

FT-IR spectroscopy was used to evaluate the coating corrosion inhibition on the surface of the Cu-based alloy. The chemical nature at the surface of the bare and coated materials were investigated by means of attenuated total reflectance Fourier-transform infrared (ATR-FTIR) spectroscopy.

The spectra were collected using a Nicolet iS50 (Thermo Fisher) spectrometer equipped with an ATR accessory. The measurements were recorded using a diamond crystal cell ATR using typically 32 scans at a resolution of 4 cm^−1^. No ATR correction has been applied to the data. The samples were investigated under the same mechanical force pushing the samples in contact with the diamond crystal.

## 4. Conclusions

The protective efficacy of chitosan-based coatings modified by the addition of cysteine was investigated to hinder degradation processes occurring in Cu-based alloys taking into account the specific requirements requested for the conservation of cultural heritage with a shiny surface typical of modern art objects.

The variation of the cysteine content has been investigated and the results reveal its key role on the inhibition effect of the active coating. In particular, our results reveal that the best efficiency in the corrosion inhibition against chloride attack increases with a minimum amount of Cy of 0.25 mg/mL due to a synergic effect between the polymer matrix and a thin layer of the amino acid. This good inhibition performance recorded for a small and simple molecule such as cysteine can be also attributed to the adsorption of the –SH group on alloy surface.

Through the FTIR analysis, it was possible to follow the interaction of Cy-alloy and Cy-CH during the accelerated test corrosion treatment.

## Figures and Tables

**Figure 1 ijms-22-10321-f001:**
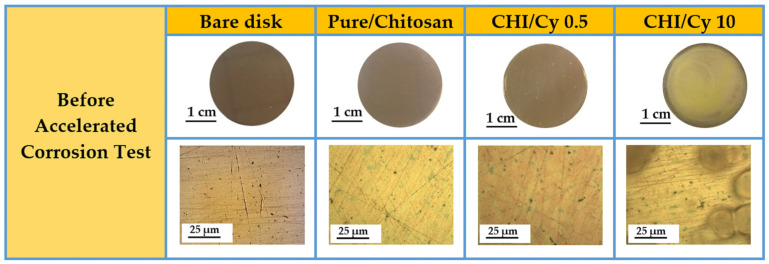
General appearance and OM micrographs of the Cu-based alloy, Cu-based alloy coated with a thin film of chitosan without inhibitor and with different concentrations of Cysteine: 0.5 mg/mL and 10 mg/mL. For all samples the scale bar corresponds to 25 µm.

**Figure 2 ijms-22-10321-f002:**
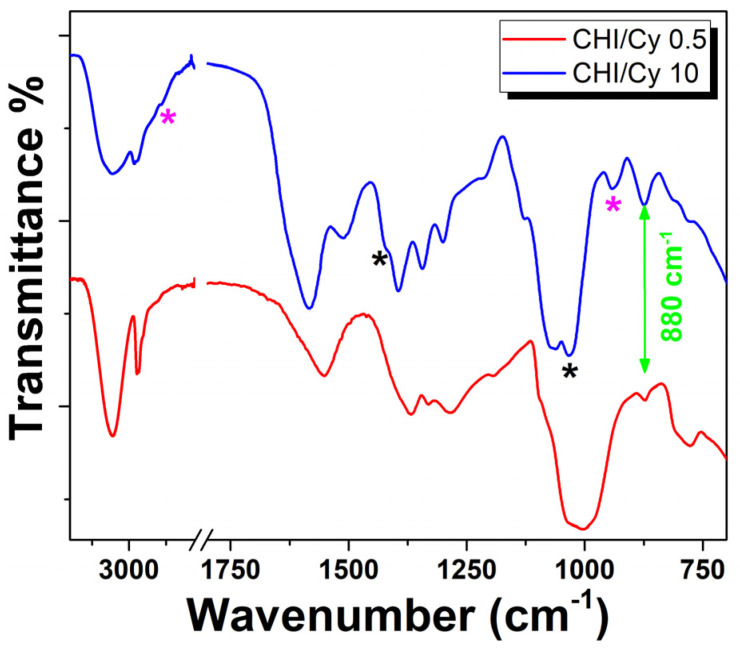
FTIR spectra of Cu-based alloy coated with a chitosan coating with different concentrations of cysteine: 0.5 mg/mL and 10 mg/mL, red and blue line, respectively. Pink stars show S-H group and black stars show aldehyde group.

**Figure 3 ijms-22-10321-f003:**
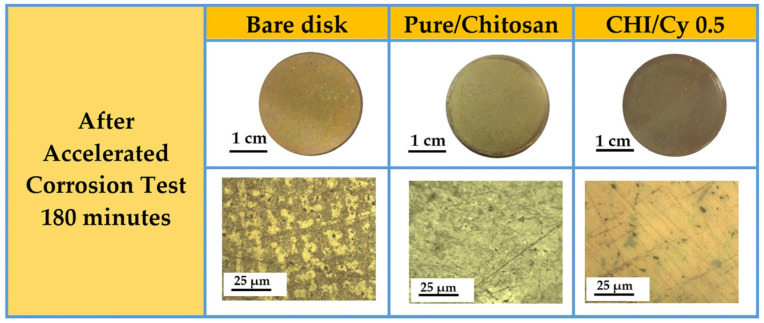
General appearance and OM micrographs of the Cu-based alloy, Cu-based alloy coated with a thin chitosan film without inhibitor and with inhibitor at concentration of cysteine: 0.5 mg/mL after accelerated corrosion test (180 min). For all samples the scale bar corresponds to 25 µm.

**Figure 4 ijms-22-10321-f004:**
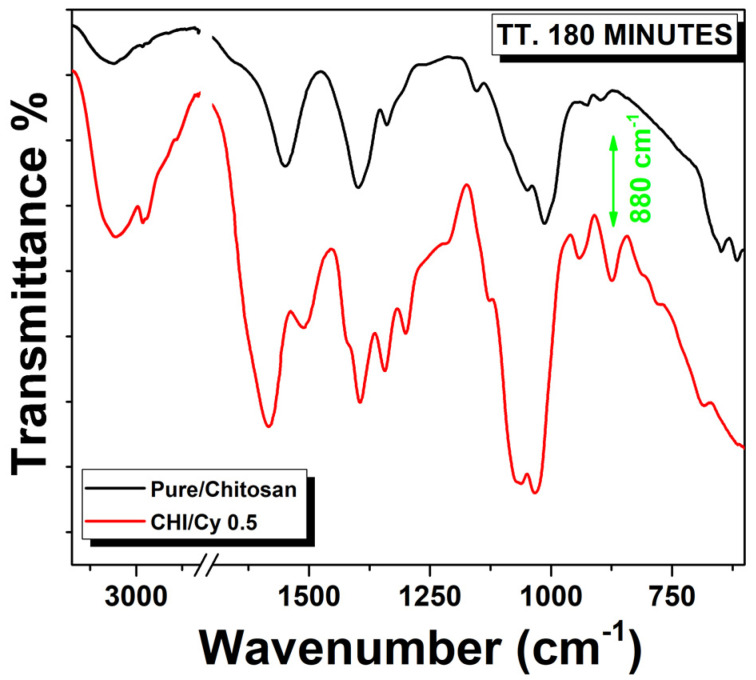
FTIR spectra of Cu-based alloy coated with a thin film of prepared chitosan film by GDL and film CHI-Cy 0.5 processed for 180 min, black and red line, respectively.

**Figure 5 ijms-22-10321-f005:**
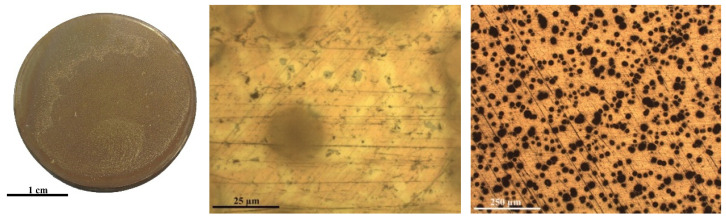
General appearance and OM micrographs of the bare Cu-based coated with a thin chitosan film with 0.5 mg/mL cysteine after an accelerated corrosion treatment was carried out for 250 min.

**Figure 6 ijms-22-10321-f006:**
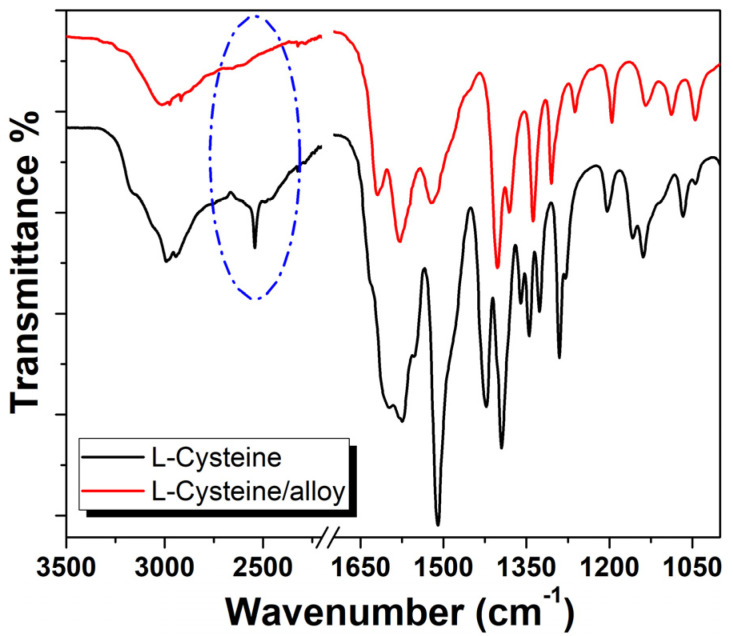
Comparison between the FTIR spectrum of cysteine deposited on the Cu-based alloy and the pure cysteine amino-acid, red and black line, respectively. Blue line show thiol vibration.

**Figure 7 ijms-22-10321-f007:**
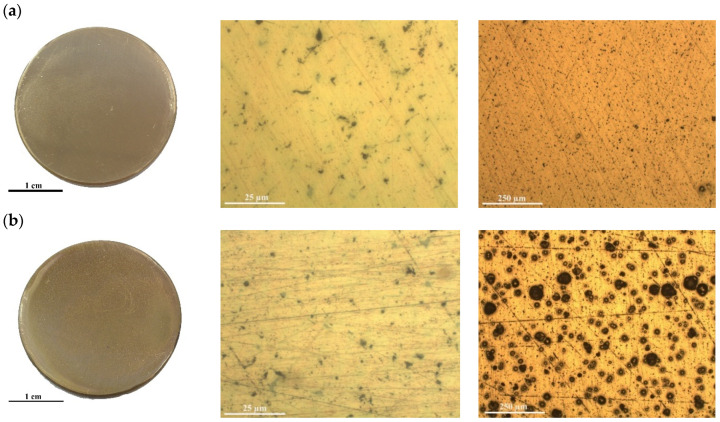
General appearance and OM micrographs of the Cu-based alloy coated with a thin chitosan film with CH/Cy 0.25 mg/mL after accelerated corrosion treatments were carried out for 180 min and 250 min, (**a**,**b**), respectively. For both examples, the scale bar corresponds to 25 µm for the high magnification image and 250 µm for the low magnification, at the center and on the right, respectively.

## Data Availability

Not applicabile.

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
