# Peer review of "Synergistic Inhibition Effect of Chitosan and L-Cysteine for the Protection of Copper-Based Alloys against Atmospheric Chloride-Induced Indoor Corrosion"

_ijms, 2021, doi:10.3390/ijms221910321_

Round 1

Reviewer 1 Report

Dear authors. In the present form this manuscript is unsuitable for publication. Major suggestions and questions are the following:

1) Such manuscript structure is absolutely unsuitable for results presentation and reading. Materials section was placed after results section? Paragraphs are too short and made text hard to understand. 

2) lines 90-92 should be deleted.

3) what was the motivation behind Cy concentration? Why 0.25? 0.5 and 10% were chosen? There is no any analysis on the selected coating parameters or composition.

4) What was the criteria of corrosion tests? If it's a surface color, then you must clearly specify the criteria of the "bad" and "good"  surface.

5) The same with the corrosion test. Why did you choose such time of the tests?

6) Overall, the manuscript is lack of data and motivation on the studied parameters. From the presented text it's hard to understand, which parameters were reference for the authors. Is it surface color or weight loss? These parameters must be specified in the text and discussed. Until that, manuscript can't be published in the presented form.

Author Response

The authors thank the reviewer for the useful comments, which have been all taken in full consideration in the revised manuscript.

Reviewer 2 Report

Check the text in lines 90-92. It is recommended to delete this text.

It is recommended to describe the applicability of the proposed anticorrosive coating for mass use.

Author Response

The authors thank the Reviewer for the positive and useful comments. As suggested a evaluation about the application on works of art was reported in the revised manuscript.

Reviewer 3 Report

Synergistic inhibition effect of Chitosan and L-Cysteine for the protection of copper-based alloys against atmospheric chloride-induced indoor corrosion, by Elena Messina, Chiara Giuliani , Marianna Pascucci , Cristina Riccucci , Maria Paola Staccioli , Monica Albini , Gabriella Di Carlo

OM micrographs need to quantify. The table with corrosion inhibition data would be welcome.

Did the authors tried to consider the surface roughness?

Did authors coatings have any biological importance?

It would be good also to discuss the problem of bacterial adhesion. I think such coatings can prevent it (Biofouling of stainless steel surfaces by four common pathogens : the effects of glucose concentration, temperature and surface roughness. Biofouling, 2019, vol. 35, 3, 273-282.).

Language and the presentation of authors work need to be improved.

Author Response

Manuscript ID: ijms-1214411

Dear Editor,

We thank the Reviewer for his/her keen interest in our manuscript and comments. The major concern of the Reviewer is the need to modify/extend the content of our paper in order to meet the readership of IJMS.

Indeed, we endorsed the Reviewer suggestions and extended our results and discussion to include a full section in the revised manuscript.

Below we report the detailed response to his/her comments.

  1. OM micrographs need to quantify. The table with corrosion inhibition data would be welcome.

As suggested by the Reviewer, a table with corrosion inhibition data was reported in the revised manuscript

  1. Did the authors tried to consider the surface roughness?

The validation on model substrates was carried out by considering different requirements that are necessary for the protection of works of art, such as transparency, removability and protective efficacy. To easily appreciate the aesthetic properties of chitosan-based coatings, the validation was carried out by using substrates with a shiny surface finish.

As suggested by the Reviewer, a critical evaluation about the application on works of art was reported in the revised manuscript

  • Did authors coatings have any biological importance?

Such measures were not provided for.

  1. It would be good also to discuss the problem of bacterial adhesion. I think such coatings can prevent it (Biofouling of stainless steel surfaces by four common pathogens: the effects of glucose concentration, temperature and surface roughness. Biofouling, 2019, vol. 35, 3, 273-282.).

Very interesting works, we will keep it in mind for future consideration.

Yours Sincerely,

The Authors

Reviewer 4 Report

Please, see attached file

Author Response

 Manuscript ID: ijms-1214411

Dear Editor,

We thank the Reviewer for his/her keen interest in our manuscript and comments. The major concern of the Reviewer is the need to modify/extend the content of our paper in order to meet the readership of IJMS.

Indeed, we endorsed the Reviewer suggestions and extended our results and discussion to include a full section in the revised manuscript.

Below we report the detailed response to his/her comments.

  1. The yellow-highlighted texts (are these introduced following possibly the reviewer comments?) very often are confusing, since they simply repeat some phrases found just before or after these texts (see, for example, the introduction section). The difficulty becomes worst, due to several typo errors and the incorrect use of English language. The authors should really try hard to give a concise a correct manuscript, in order to avoid confusing the reader.
  1. Experimental section: The coating method should be described. More, important the composition of the coatings is not clear. A table compiling all coating attempts and composition of the coating solutions is necessary

As suggested by the Reviewer, a table with corrosion inhibition data was reported in the revised manuscript

  • Optical microcopy images. To evaluate anticorrosion efficiency, the authors use just optical microcopy images. There is no attempt to somehow quantify this efficiency. In most figures showing optical microscopy images, the evolution with the treatment time is reported. However, it would be much clearer to compare the optical microscopy, before and after accelerated corrosion treatment, of uncoated substrate, substrate coated with pure chitosan and substrate coated with several contents of cysteine. This order of presentation is more rational and would make the discussion clearer.

As suggested by the Reviewer, the text was modificated in the revised manuscript

  1. ATR-FTIR spectra. The same holds for the ATR-FTIR presentation and discussion. In this case, the presentation of the chemical structures of all components of the coatings is necessary. As far as I understand the coatings contain a large amount of gluconolactone (or gluconic acid/salt, if hydrolyzed). Have the authors taken into account this component when attributing the spectra?

All spectra taken into account the peak of GDL and Chitosan according to data reported in previous work (Progr. Org. Coat. 2018, 122, pp. 138–146.)

  1. Page 6, lines 220-222. I really can not understand this phrase Page 6, line 229: “(mainly copper oxide and copper hydroxychlorides) can be recognized”. Could you, please, give more details?

These are corrosion products more prevalent in copper based alloys. But in our system, due to low concentration are hardly observed with FTIR analysis. 

  1. Figure 6. What is the anticorrosion efficiency of the substrate treated with 0.02 mg/ml cysteine? In view of all these remarks, a major revision is at least needed before reconsideration of the present manuscript.

As suggested by the Reviewer, the text  was modificated in the revised manuscript

Yours Sincerely,

The Authors

Round 2

Reviewer 1 Report

Dear Authors, such structure of the manuscript is not suitable for publication as I pointed in the previous review. Results and Discussion section must be presented after Materials and methods. A lot of paragraphs are consisted only from one sentence and is hard to read and understand.

Besides that, questions from Review 1 were not addressed:

1) Why  Cy concentration of 0.25 mg/ml is the minimum amount? It seems that it's minimum amount that you have tried. But maybe 0.1 mg/ml will be with the same corrosion results. This statement need to be additionally discussed or edited and supported by the results.

2) The same with corrosion test time. Why 250 min is long enough time? Maybe there will be changes after 300 minutes.

3) It's still not clear what was the criteria of the corrosion tests. If the corrosion products then which one? If only corrosion products then what is the reason behind OM analysis? Please, explain it in the methods section.

Please, provide the described above data to support the results and conclusions. 

Author Response

Dear Editor,

We thank the Reviewer for his/her keen interest in our manuscript and comments. The major concern of the Reviewer is the need to modify/extend the content of our paper in order to meet the readership of IJMS.

Indeed we endorsed the Reviewer suggestions and extended our results and discussion to include a full section (Section... in the revised manuscript) on Cy concentration

Below we report the detailed response to his/her comments.

  1. Results and Discussion section must be presented after Materials and methods. A lot of paragraphs are consisted only from one sentence and is hard to read and understand.

As, We have explained in the previous review a template was used to write the manuscript. In this template requests to submit Results and Discussion before the Materials and Methods. Should we consider asking to publisher to reverse section so that the reading it's easy

  1. Why Cy concentration of 0.25 mg/ml is the minimum amount? It seems that it's minimum amount that you have tried. But maybe 0.1 mg/ml will be with the same corrosion results. This statement need to be additionally discussed or edited and supported by the results.

As, we have explained in the previous review the inhibitor concentration has been chosen on the basis of literature data. This data revealed (rif. 23) that CY is a good inhibitor for copper corrosion at about 18 mM in HCl solution.

The minimum concentration of amino acid (CY) that has been applied in this study was 0.25 mg/ml. (i.e. 2 mM), as the use of chitosan which act as barrier layer permit for further lowering amino acid concentration.

We thank the Reviewer for pointing this out. Indeed we have no conducted experiments at 0.1 mg/ml, so we cannot say taht the amino acid performs similarly.

  • The same with corrosion test time. Why 250 min is long enough time? Maybe there will be changes after 300 minutes.

As reported in revisited manuscript already in 180 minutes the occurrence of corrosion processes was clearly observed in the bare disk. Then 250 minutes can be considered a sufficient time to observed the properties of chitosan based films as a inhibitor corrosion.

  1. It's still not clear what was the criteria of the corrosion tests. If the corrosion products then which one? If only corrosion products then what is the reason behind OM analysis? Please, explain it in the methods section.

As suggested by the Reviewer, a more detailed description about the methodology explanation was reported in the revised manuscript.

Yours Sincerely,

The Authors

Reviewer 3 Report

Not all comments were taken into account!

Author Response

Manuscript ID: ijms-1214411

Dear Editor,

We thank the Reviewer for his/her keen interest in our manuscript and comments. The major concern of the Reviewer is the need to modify/extend the content of our paper in order to meet the readership of IJMS.

Below we report the detailed response to his/her comments.

  1. Not all comments were taken into account!

I blame my self for not having been more clear in previous revision.

To easily appreciate the properties of chitosan-based coatings, the validation tests was carried out by using substrates with a shiny surface finish. The results of the image analysis, performed to estimate the percentage of corroded surface, are reported in revisited manuscript. These confirm that chitosan/inhibitor coatings noticeably protect the bronze surface from corrosion and their efficacy is much more pronounced in the presence of 0.25 mg/ml Cy. In  particular, after 180 minutes of accelerated corrosion treatment the surface bare alloy is totally corroded, unlike cannot be said  of surface in the presence of chitosan/Cy.

The authors thanks the Reviewer for his/her suggestion, regarding the problem of bacterial adhesion and the possibility to use coatings can prevent it. We will keep it in mind for future consideration.

Reviewer 4 Report

The authors have put considerable efforts to ameliorate the quality of the presentation of this manuscript.

The manuscript can be accepted for publication. However, the use of English language should be checked, especially in the highlighted texts.

Author Response

Manuscript ID: ijms-1214411

Dear Editor,

We thank the Reviewer for his/her keen interest in our manuscript and comments. The major concern of the Reviewer is the need to modify/extend the content of our paper in order to meet the readership of IJMS.

Below we report the detailed response to his/her comments.

  1. The authors have put considerable efforts to ameliorate the quality of the presentation of this manuscript. The manuscript can be accepted for publication. However, the use of English language should be checked, especially in the highlighted texts.

The authors thanks the Reviewer for his/her positive and useful comments. According to the Reviewers suggestion, the use of English language should be checked.

Round 3

Reviewer 1 Report

I see some improvements of the manuscript and can recommend it for publication. But i still suggest to place Materials and Methods section before results as it will make manuscript more clear.

Author Response

Manuscript ID: ijms-1214411

Dear Editor,

We thank the Reviewer for his/her keen interest in our manuscript and comments. The major concern of the Reviewer is the need to modify/extend the content of our paper in order to meet the readership of IJMS.

Below we report the detailed response to his/her comments.

  1. I see some improvements of the manuscript and can recommend it for publication. But i still suggest to place Materials and Methods section before results as it will make manuscript more clear.

The Authors thanks the Reviewer for his/her positive and useful comments. According to the Reviewers suggestion, we asked to editor to reverse section so that the reading it's easy

Reviewer 3 Report

I do not see the corrections in the last version.

Here are my additional comments:

How large are experimental error and how reliable are the results.

The author needs to check the adsorption properties of Chitosan. How good is covered the surface? I propose to used AFM or detailed SEM.

Author Response

Manuscript ID: ijms-1214411

Dear Editor,

we thank the Reviewer for his/her keen interest in our manuscript and comments. The major concern of the Reviewer is the need to modify/extend the content of our paper in order to meet the readership of IJMS.

Below we report the detailed response to his/her comments.

  1. I do not see the corrections in the last version.

I’m sorry, if in the latest version I forgot to underline in yellow the editings.

  1. How large are experimental error and how reliable are the results.

Five independent measurements were carried out for each type of coating investigated to take into account the variability due to the surface of the bronze disks. In such a way, the reproducibility of the CH/Cy 0.25 mg/ml was verified as a best coating.

  • The author needs to check the adsorption properties of Chitosan. How good is covered the surface? I propose to used AFM or detailed SEM. 

The chitosan films appears colourless, more transparent and uniform by Optical Images analysis as a reported in the manuscript.

In order to get more information about their formation, chitosan films was also characterized by SEM microscopy. It is worth noting that SEM analyses show uniform and cracks-free films, the same we can be observed by OM. For this reason have not been reported in the manuscript

Round 4

Reviewer 3 Report

Not all comments were take into account